# Ineffable absences, irrefutable presences

Gisela Heffes

Department of Modern Languages and Literatures, Johns Hopkins University, Baltimore, MA, USA

## Perspective

de-extinction; rewilding; material objects; creative writing; futurities

**Corresponding author:**
Gisela Heffes;
Email: gisela.heffes@jhu.edu

I am grateful to the two anonymous reviewers whose insightful feedback helped me improve this piece. I was fortunate to receive suggestions that not only enhanced my article but also prompted me to reconsider these observations and notes one more time, through their input.

## Abstract

This speculative essay examines the concepts of extinction and (de)extinction through the lens of quotidian objects, emphasizing that each material artifact tells a story about its ingrained elements and the "absence" it signifies. Situated within the framework of the Anthropocene, this reflection draws inspiration from a recent exhibit at the Peale Museum, showcasing artifacts retrieved from sites along the Jones Falls River and the Chesapeake Bay in Baltimore, MD. Adopting an interdisciplinary approach that contemplates futuristic visions of place and the embedded histories within objects, such as dolls, calculators and soda cans, the essay envisions a Museum of Extinction that interrogates the stark contrasts between tangible remnants of the natural world and living organisms in constructed environments. These objects embody haunting stories of damaging extractive practices and ecological and cultural erasure, serving as poignant reminders of the subtle presence of vanished lives and species, compelling us to deepen our understanding of the intricate dynamics of production, consumption and loss. It contends that, although a single or definitive "formula" for de-extinction is unattainable, poetic and creative engagements with everyday artifacts can serve as powerful testimonials to absences and material interventions. Such acts of writing not only foster a profound understanding of ecological and cultural entanglements but also motivate active material interventions. They transform the act of writing about objects into a reflective practice – an invocation of remembrance and a catalyst for potential change, sparking a quiet insurrection in how we perceive and care for the world.

## Impact statement

This speculative essay underlines the relevance of creative writing, storytelling and poetic undertakings that engage with the material artifacts that accompany our daily lives, aiming to gain a deeper understanding of extinction. It advocates exploring the absences left by extinction through creative endeavors, inviting readers to reflect on and meditate about the interplay between what is lost and how it may transform us in the present, directing us toward the future. By imagining futuristic visions of place, it envisions what could be displayed in a Museum of Extinction, in a posthuman world.

## Ineffable absences

Can we gain a deeper understanding of extinction by examining the materiality of the objects around us? What do we observe when we examine a doll, a penny or a rusty can of Coca-Cola more closely? What secrets are concealed behind the wrinkles of a Goodyear tire? Can we evaluate the scale and scope of extinction through the lens of everyday items, such as a spatula or a notepad? In this essay, I build on the notion that material artifacts serve as biographies of the elements that make them, revealing the "absence" left by the extracted materials in their environments. However, this is not just a single biography; rather, it is a culmination of organic and inorganic lives that have fossilized over time, producing the material elements that constitute the essence of these artifacts. This speculative essay does not aim to propose a formula for de-extinction. What has been lost cannot be regained. However, one may ask if a species is revived, what remains absent? Or, more specifically, what remains? Even though species could be "revived" (as the Colossal Bioscience Tech company aims to do), what remains absent is their existence prior to extinction. That is, the memory of something that once existed and became extinct, like the long tail of gas and dust left by stars, or an imperceptible imprint of devastation that was not only unnecessary but also avoidable. And within that trace, what remains absent is that imperceptible record, an encapsulated imprint that warns us about the destructive actions of human beings.[1] Nevertheless, we can renew our perspective to recognize that

---

[1] Joanna Zylinska notes, in *Nonhuman Photography*, that the "de-extinctionist mode of thinking" conceives of "organisms as individual entities that can simply be reinserted into various environments, rather than as mutually constituted with them" therefore sliding over the "issue of our human responsibility toward the biosphere by focusing on singular successes of the survival and revival of 'charismatic megafauna': the useful and the cute" (2017, 99). In addition, many critics have raised concerns about the feasibility, ecological risks, ethical issues, and resource allocation, suggesting that de-extinction might distract from efforts to save existing species. Furthermore, some scientists argue that some of these outcomes are genetically engineered proxies rather than true de-extinctions, and that ecosystem contexts have changed. See, for instance, Patrick Greenfield's piece "It's 12 ft. tall, covered in feathers and has been extinct for 600 years – can the giant moa bird really be resurrected?" (2025).

our daily material experiences remind us of these absences. Ultimately, the haunting stories embodied by these elements can inspire us to gain a deeper understanding of the complex relationship between production, consumption and extinction. And perhaps, slow down its pace.

### The future of us

Two months ago, I was invited to give an informal talk based on a collaborative exhibition at the Peale Museum in Baltimore titled *The Future of Here: A Glimpse of a River Culture to Come* (February 13, 2025–March 30, 2025). The exhibition is designed as "an invitation to reimagine our place along the Jones Falls River and the Chesapeake Bay in a distant future beyond our fossil-fueled present"; it showcases works from a group of artists and researchers at Johns Hopkins University, stemming from a course co-led by visual artist Jordan Tierney and environmental anthropologist Anand Pandian in the fall of 2024, and displays artifacts and insights from a shared exploration of the future.[2]

When I read the question that prompted the discussion, "What artifacts might the people of that future time produce, and how might they creatively use the many things we leave behind?" I immediately thought of my own interests in the stories embedded in the materiality of objects and the prospect of artifacts that might survive us as humans, potentially displayed in a Museum of Extinction.[3] However, rather than considering the extinction of other nonhuman organisms, I reflected on the question from the reverse perspective, viewing it as concerning us, the humans, who may become extinct.

If the first question is articulated around the idea of the artifacts, it also inquires about how we can envision what these places have been and what they may yet become as elements of future lives. These are questions about objects and places confronting the future, or perhaps a future, if such a possibility remains tangible. But what about our future? This is not a trivial question. One might consider, for instance, these objects and places as part of a posthuman world, as the recently acclaimed film *Flow* (2024) by Latvian director Gints Zilbalodis illustrates.[4] The animated film unfolds in a world inhabited solely by nonhuman beings, where natural disasters have become a frequent and ordinary occurrence, and references to the remnants of human civilizations imply the possibility of human extinction.

In some ways, I thought this prompt could provide an opportunity to explore collective ways of poeticizing environmental collapse and to examine a series of dilemmas, such as how to narrate the climate crisis, which imaginaries we turn to and what forms of representation we employ. Equally important, however, is the beautifully posed question of "to reforest our own imaginations, thinking and working through the ruins of our time?"[5] This led me to consider rewilding not only our environments but also how to rewild our own perspectives. The exhibition showcased a series of artifacts found at the site along the river, which were collected and displayed in a glass box. I view these objects as relics that not only will outlast us but, more importantly, will tell their own stories.[6] The artifacts displayed in the exhibition have the ability to represent our current era, considering the human imprint on the planet and its vestiges in the natural world. They connect us to the past. A calculator, for instance, may have been used by a child learning math and preparing for college entrance examinations. The plastic half-arm with its small hand from a broken doll evokes childhood, joy and affection. The flat aluminum from various beverage cans, such as Pepsi or Red Bull, encapsulates the ephemeral nature of quenching thirst, made easily accessible by the tin can sold at a 7-Eleven convenience store. At the same time, ironically, the materiality of its purpose will endure for an extended period. The plastic and tin abandoned in these natural sites are reminiscent of their extractive ancestry. Moreover, they remind us that the geology of the Earth, the layers that compose its strata, is shaped by the materiality of our desires, needs and purposes in life. Although these desires, needs and purposes are transient, their impact will be permanent.

The artifacts displayed in the glass box, which may not connect to the place where they were found, populate the space in a surprising and possibly unrelated manner. As they crowd the territories designed to showcase our dreams, their existence serves as a corroboration of what has ceased to exist. Their presence correlates with a myriad of extinctions. One might ask, for instance, what does the existence of this doll, calculator or Pepsi erase, and how is its presence related to another absence? What questions do these artifacts raise about the future? Do they evoke buried emotions? What variety and scope of earthly meanings and muddy stories can they encompass? By extinction, I mean what has been extracted to make the doll, the calculator and the Pepsi possible. Their presence invites us to reflect on the sacrifice and disposal zones whose imprint is rarely made visible to us as we look at the ruined artifacts of the present – evident in our daily itineraries through the current landscapes. While these artifacts weave a network of voices embedded in both their materiality and immateriality, creating a polyphonic fabric where extinctions and presences overlap, they also point toward a future.

### (De)extinction and (re)wilding

I am always intrigued by what the reverse side of things, stories and lives can reveal. There is often a hidden, or perhaps less visible, narrative that tells a different story from what it was intended to convey. An artifact serves as an archive of a wide range of stories that converge within the materials assembled to create such an object. As geologist Marcia Bjornerud (2024) explains in a beautiful

---

[2]For more information on the exhibition, go to https://thepeale.org/exhibition-future-is-here/.

[3]Unlike the museums that surround us today, a future Museum of Extinction would be organized – if such a word can be used – according to a posthuman arrangement where, perhaps, if humans were also to become extinct, artifacts would be displayed in a non-anthropocentric manner, entangling objects beyond human-centric frames and therefore beyond conventional dispositions and taxonomies.

[4]*Flow* (2024), directed by Gints Zilbalodis, written and produced by Zilbalodis and Matīss Kaža.

[5]The invitation came from Johns Hopkins University anthropologist Anand Pandian to a group of scholars working on ecological humanities and across departments and disciplines.

[6]By "tell their own stories," I do not mean "speak," but rather to communicate. Drawing from the concept of material agency as defined and developed by Bruno Latour (2005) and Jane Bennett (2010), I propose a view of communication connected to the idea of biosemiotics (Barbieri, 2009). By articulating these two approaches, I see materiality as a sign carrier, understanding matter not just as a passive medium for signs but as an active agent that shapes and constrains meaning. Furthermore, if biosemiotics cannot be separated from the material world but, instead, is deeply intertwined with and emerges from the processes of life itself (Wheeler, 2016), then Bennett's "vibrant matter" can be read as a semiotic actor that influences perception, emotion, and behavior. From this perspective, objects not only communicate but are also polyphonic (in the sense that their significance and meaning unfold through multiple and varied perceptions), and their substance is fluid – not fixed.

piece titled "Wrinkled Time: The Persistence of Past Worlds on Earth," an "exceptional yet less obvious attribute is the way that Earth preserves accessible records of countless earlier versions of itself, condensing events of billions of years into the volume of the present-day continents."[7] In achieving this feat of compression, Earth effectively wrinkles time – accordioning eons and juxtaposing moments in its long history – through the medium of rocks.[8] Similar to the mysterious stories archived in the rocks, man-made objects store a myriad of tales embedded in the wrinkles described by Bjornerud. However, when they are found in places such as forests, rivers and beaches, we deem them out of place – an "externality" that belongs to another environment and therefore feels inadequate. Their contaminating presence undermines the beauty and "pristine" quality of the supposedly unspoiled site. But what about the reverse side of nature? What about urban, artificial and nonnatural places?

Thinking of (re)wilding and de(extinction) forces us to examine our built environments as well. Challenging the assumption that there is nothing natural in the cities we inhabit, I found glimmers of life and hope in these seemingly hopeless spaces. Cities both allow and offer us a perspective on nature from a completely different angle. In the forest, we see the whole. We perceive fragments of objects as remnants of a postnatural world, intervened by alien and inorganic elements. In the city, however, this constructed and artificial whole is shattered by the sprouts of nature that push to emerge. Shoots of weeds and tiny flowers resist the burden of asphalt, questioning the imposed fate of dying, disappearing and becoming extinct. On the sidewalks, tree roots expand underground, creating cracks and grooves that disrupt the paving stones. These fissures exclaim, "Here I am, I exist." Tiny shoots that defy the human dictum that weighs down and seeks to suffocate them. It is these bursts of life amidst human artificiality that sustain the promise of a rewilding that is possible, beyond human agency.

### Writing against extinction

Art can serve as an antidote against obliteration. For those of us haunted by the ghosts of extinction, or by extinct beings becoming ghostly presences, writing can function not only as a tool to formulate questions and inquire about our surroundings but also to seek a better understanding, give meaning and make sense of the world around us. Although seemingly immaterial, a ghost can be a voice, an image, a fear or something that haunts us – a latent presence struggling to emerge, inciting us to abandon all restraint and turn to creative forms of expression. When contemplating objects, stories and artifacts in the wilderness and the wild within artificial environments, I turn to this "ghost" because it compels me to accept, identify and transform an anxious concern into storytelling. Rather than fleeing from these fears, no matter how painful they may be, I find it more effective to confront them. To reverse the rules and, instead of becoming paralyzed in the face of its devastating presence, to act: to seek them out, pursue them and materialize their incorporeality into textual substance – to create a fabric of words that exorcize these ghosts in a liberating act. Furthermore,

capturing them in the texture of language will enable us to materialize the potential threat of becoming nonexistent.

Writing is a profound act of intervention – an instrumental means to traverse and explore the spectralities that, whether overt or covert, converge, intersect and intertwine with the current ecological crisis. These spectralities, even in their presumed immateriality, can materialize tangibly through the presence of objects, rendering the unseen perceptible. Rather than mere contemplative gestures, writing embodies a tangible, corporeal engagement – a palpable intervention within the fabric of the world. It acts as a conduit through which knowledge, both ancestral and contemporary, imaginaries, cultural and personal archives, stories, aesthetic preferences, communicative predilections, anxieties and hope flow and entangle. Moreover, writing is an inherently material practice, an organic activity with which we engage. As the visual artist Christopher Volpe (2020) eloquently asserts, art compels us to act despite the ominous shadow of "humanity's looming demise."[9] This is because "We treasure, in the strongest art, authentic and profoundly meaningful intuitions of the conditions of being human," which makes "this any clearer or more important."[10] And, in doing so, it offers a stark reflection of our shared existence and vulnerabilities amidst the specter of ecological collapse.

One might wonder why we should pay attention to objects when the Anthropocene encourages us to think about the crisis across geological eras and on a planetary scale. Perhaps, as Neil MacGregor suggests in *A History of the World in 100 Objects* (2010), the narrative conveyed through objects reflects the stories of entire societies and complex processes rather than individual events. If a casual observer in the future were to see the objects collected at the sites along the river, or in places such as forests, desserts and beaches where they do not belong, would they be able to hear the stories that emanate from their materiality? What would they think? Would they recognize them? What emotions would they feel? These relics account for what was extinguished, what was extracted and what was created from the physical removal process of matter.

Writing about these objects encourages exploration that enables the experience of various ecological crises, the possibility of hope and the probabilities for an alternative future. By tuning into the bodily senses, the objects trigger sensory itineraries that may combine to create synesthesia: a register of how extinction is felt and what its absences and presences signify. From articulations of difficulty to what is challenging to articulate, the materiality of objects converges with public spaces and institutions, memories embodied in monuments and transitory aspects, such as residues and ruins. Writing a journey toward the future represents a transition between scales, shifting from the materiality of the monumental to smaller, local objects that form a little thread or tangle, allowing us to glimpse the connections and fabric of a new future on Earth.

Each object functions within its own sphere, supported by a backbone that can be interpreted through a juxtaposed reading where the historical materiality of its layers nests and extends beyond personal affections. These layers connect to the deep time of the Anthropocene. These elemental compositions – the history of

[7]Bjornerud, Marcia. "Wrinked Time: the Persistence of past Worlds on Earth:" *Emergence Magazine* (October 17, 2024): Accessed April 4, 2025: https://emergencemagazine.org/essay/wrinkled-time/.

[8]Bjornerud, Marcia. "Wrinked Time: the Persistence of past Worlds on Earth:" *Emergence Magazine* (October 17, 2024): Accessed April 4, 2025: https://emergencemagazine.org/essay/wrinkled-time/.

[9]Christopher Volpe. "On Making Art in the Anthropocene:" *Millennium Alliance for Humanity and the Biosphere* (MAHB) (April 23, 2020): Accessed July 13, 2025: https://mahb.stanford.edu/blog/on-making-art-in-the-anthropocene/.

[10]Christopher Volpe. "On Making Art in the Anthropocene:" *Millennium Alliance for Humanity and the Biosphere* (MAHB) (April 23, 2020): Accessed July 13, 2025: https://mahb.stanford.edu/blog/on-making-art-in-the-anthropocene/.

materiality itself – trigger a multitude of pasts. In this sense, they confer a certain ambiguity to the object, as on one hand, there are both the sensory and affective experiences that the object invokes; on the other hand, their very materiality denaturalizes it. This denaturalization challenges a presumed purity – similar to that of the calculator or the plastic doll – and removes it from the realm of the anodyne. The materiality of the objects engages in a relational dialog through two perspectives: one transversal and one perpendicular. This approach fosters a deeper understanding of the distant past and the transformations we are undergoing, questioning certain assumptions tied to the idea of the uncontaminated and the everyday.

### Future perspectives or perspectives of/for the future

The Bolivian thinker Silvia Rivera Cusicanqui (2015) writes that in Aymara culture, the future is behind us and the past is in front: we can see what we know, but we cannot see the unknown. Imagining what places have been and what they may yet become as components of future lives means embracing Cusicanqui's proposal of "carrying the future on their shoulders and looking at the past with their eyes" (211; my translation).[11] This involves connecting these relics of our time to the natural and cultural vitality of other places. Reflecting on and working through the ruins of our time requires both an exercise of imagination and situating our gaze at a different angle. If we can "reforest our imagination," we will certainly be able to reforest our immediate and distant environments.

Is it possible to capture what is dissipating? Be it a butterfly, a mountain or the blue ocean, as we experienced in childhood, read about in books or envisioned through the oral stories recounted by grandparents, uncles, teachers, friends or strangers? How do we document extinction? How can we narrate it? How can we claim it before it vanishes amidst the rapid changes in records? Art has the potential to redress what would otherwise be lost. Art can help to de-extinct our capacity to transform ourselves and others, allowing us to germinate in unprecedented ways. The artifacts evoked in this piece not only serve as a testament to such absences; more importantly, their undeniable presence reverts the ineffable absences that lie behind their shapes, smells, textures and colors. Throughout history, artists have grappled with the concept of absence, exploring the silent spaces where meaning lingers. Interested in how artists envision to portray "the absence of a thing," curator Gianni Jetzer organized the exhibition "What Absence Is Made Of" at the Hirshhorn Museum in Washington, DC.[12] This exploration encompassed not only the depiction of tangible voids, such as the gaping hole left in a heart by tragic loss, but also the elusive presence of invisible absences that subtly shape the fabric of our physical reality. Art memorializes what has vanished, yet it also foregrounds the persistent presence of extinction's silent absence within our collective consciousness. Although confronting the profound challenge of redressing losses wrought by extinction remains daunting, artistic expression becomes a vital conduit for confronting grief, especially regarding the vanishing of species and habitats. Such memorialization does not imply restitution; instead, it creates sacred spaces for remembrance and mourning. In this manner, art emerges as a poignant testament – an enduring testament – to the voids left behind, ensuring that these profound absences are woven into the ongoing narrative of human and ecological memory.

### Irrefutable presences

As a writer and scholar, I propose to explore the relationship between the materiality of artifacts and the materiality of writing. Within its lyrical potential lies the possibility of poetizing "the ruins of our time," highlighting that the materiality of those objects influences us, and we, in turn, influence that matter. The relevance of storytelling in recognizing the wrecks of our era stands out as a powerful and recurring motif across various academic disciplines. Works such as *Arts of Living on a Damaged Planet* (Tsing et al., 2017) evoke ghosts and monsters as metaphors for grappling with themes of extinction, decay and resilience – narratives that serve as vital pathways to living meaningfully amid a fractured Earth. Similarly, renowned biologist Donna Haraway (2016) advocates for "speculative fabulation," a storytelling paradigm that resists the despair of dystopian narratives and cultivates visions of collaborative multispecies futures. Literary scholar Ursula K. Heise, in *Imagining Extinction* (2016), explores how cultural narratives – particularly in speculative fiction and biodiversity discussions – foster society's engagement with extinction, inspiring both reflection and action through ethical frameworks.

Writing, and, more broadly, art, allows us to collectively explore, experiment and establish the conditions necessary for a situated intervention to become attainable. If the current relics scattered throughout natural environments represent what was extinguished, what was extracted and what was generated in its place, it is also through registering their absence that we might collectively compose an archive of extinction – one that lays the foundation for both a human and nonhuman animated world, and one that serves as an intervention for other modes of potentially de-extinct. For through the crevices of ruination, the vital pulse of life never stops flowing.

**Open peer review.** To view the open peer review materials for this article, please visit http://doi.org/10.1017/ext.2025.10004.

**Acknowledgements.** I am thankful to Anand Pandian for the initial invitation to participate in a public discussion about the exhibition *The Future of Here: A Glimpse of a River Culture to Come* (February 13, 2025–March 30, 2025), co-curated with artist Jordan Tierney at The Peale in Baltimore, MD and supported by the Krieger School of Arts & Sciences at Johns Hopkins University, the JHU Department of Anthropology and the JHU Program in Museums & Society, the Alexander Grass Humanities Institute and the Ecological Design Collective. Reflecting on the remains displayed in the exhibition sparked the idea for developing this essay further.

**Author contribution.** The author conceived and drafted the piece in full.

**Competing interests.** The author declares none.

---

[11]"el aquí-ahora de la historia, el espacio-tiempo en el que la sociedad 'camina' por su senda, cargando el futuro en sus espaldas (*qhipha*) y mirando el pasado con los ojos (*nayra*), como lo dice el proverbio *Qhip nayr uñtasis sarnaqapxañani.*"

[12]Brendan L. Smith. "Absence in art and the presence it creates at the Hirshhorn." *Smithsonian Insider* (December 27, 2017): Accessed July 12, 2025: https://insider.si.edu/2017/12/absence-art-presence-creates-hirsh horn/#:~:text=The%20effects%20of%20our%20digitallyHirshhorn%20Museum %20through%20summer%202019.

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
