## [Reviewer Report]

Though I am in the humanities, I am a philosopher. So, my comments reflect a more philosophical bent in reading this essay. Here are some thoughts and questions. I am inclined to ask for revisions in part to address questions I raise but to invite further exposition of the author’s aims in the essay.

The author explores how writing, and the materiality of writing, and writing about material things can help alert us to what has been lost through extinction. In effect, we can explore the absences of extinction through this creative act. This piece of writing itself is a meditation on remembering what is lost and how that may transform us in the present in directing ourselves to the future.

• The author writes that, “What has been lost cannot be regained.” Those at Colossal Biosciences would of course disagree. So, I wonder if a species is revived, what remains absent?

• They also mention objects and artifacts displayed in a future “Museum of Extinction.” But are we not already surrounded by these museums? Does the fact that we continue to dominate our planet tell us something about our future? Are we misunderstanding what these museums are?

• I agree that, “Art can serve as an antidote against extinction.” But it seems from the essay that the author thinks that this is mostly in making sense of our world. Do they think that this is mostly a contemplative activity or one of action?

• I was struck by the sentence, “Art has the potential to redress what would otherwise be lost.” One way to understand redress is to make someone harmed whole. It is true that art can transform us, but how can it redress those lost through extinction? Learning to recognize “their undeniable presence” is of great importance but I wonder if that is enough.

---

## [Reviewer Report]

This provocative and poetic reflection on the material culture of the Anthropocene takes its starting point from a recent exhibit at the Peale Museum, exploring futuristic visions of place “beyond in the area around Chesapeake Bay and its environs, an artifact of interdisciplinary collaborations with the arts. Coming from an environmental humanities perspective, it imagines what might be displayed in a “museum of extinction” along Chesapeake Bay. Exploring the idea of accounts that could be told about our harmful extractive activities by the histories inscribed in everyday objects (a doll, a calculator, a soft drink can), it considers what might be leftover from the current era, the presence of which also signals the absences of lost lives and lost species. The haunting stories embodied by these items can inspire us to “better understand the relationship between production, consumption, and extinction” (p.1).

The value of storytelling to recognize “the ruins of our time” (p.8) is a theme that has recurred in a number of disciplines, and there could be a little more grounding in relevant scholarly literature without overburdening the narrative. Since you mention environmental anthropologist Anand Pandian’s significant role in creation of the museum feature, it might also be useful to note some of his writing, as well as Anna L. Tsing’s well-known work, The Mushroom at the End of the World: On the Possibility of Life in Capitalist Ruins (Princeton University Press, 2015), and also engage with Gaston R. Gordillo’s evocative book, Rubble: The Afterlife of Destruction (Duke University Press, 2014).

I have some hesitation over the cultural assumptions implied by the move to take for granted that the objects in the exhibit have the capacity to speak for themselves. You explain, “I view these objects as relics that not only will outlast us but, more importantly, will tell their own stories” (p.3). I would say, rather, that the semiotic power of such objects is that which is given to them by the beholder, or perhaps, by the historically shaped, cultural perspectives engendered among a community of beholders. A whole kaleidoscope of possible stories or fragments of stories might be embedded in material objects, and there is no monolithic, predetermined way that such stories are necessarily “read”. Rather, the significance of objects unfolds through our conversations with them and each other, at particular moments in time. While today, in the context of a provocative art exhibit, I might be inspired to recognize the history of a particular soft drink company’s water mining in an underdeveloped periphery, tomorrow, in the midst of a heatwave, I might simply be raked with thirst. Or, following June Nash’s wonderful article, “Consuming Interests” (Cultural Anthropology, 2007), someone from Chiapas could associate it with a long tradition of Catholic and Indigenous ritual. While another person might consider the flavour too alien and vile to pause and think about it at all. So, I would suggest bracketing the discussion with a recognition that the signs we see are inherently polyphonic, and their significance is fluid.

Because the visual arts are so instrumental to the discussion, I would suggest citing the link to the museum’s site, where selected photographs might remain archived:

https://thepeale.org/exhibition-future-is-here/

Thank you so much for awaiting my review; in a period filled with constant administrative burdens, it was a pleasure to think about such an engaging essay, with its multidisciplinary resonance. Please note that there is a missing end of sentence on p. 4; there are occasional gaps in editing (for example, “shapes, smells, and textures colors”, p. 8) and Anand Pandian’s name is mispelled in the notes. Otherwise, the piece is written with a mature and skilled voice.

---

## [Editor Report]

Dear Gisela,

I trust you are well. Thank you for submitting this thought-provoking manuscript that has been positively viewed by the reviewers. The reviewers raise some interesting points that I do not believe will be difficult to address, hence I recommend minor corrections.

Best Wishes

Dave

---

## [Editor Report]

Thank you for your submission and excellent revisions, they were easy to follow and addressed the reviewers' comments. I am therefore happy to recommend the manuscript be accepted. Being from the biology side, I found this a fascinating read and gave me lots of food for thought. I look forward to seeing it published and sharing it with colleagues.